# Impact of Various Catalysts on Transesterification of Used Cooking Oil and Foaming Processes of Polyurethane Systems

**DOI:** 10.3390/ma15217807

**Published:** 2022-11-05

**Authors:** Elżbieta Malewska, Krzysztof Polaczek, Maria Kurańska

**Affiliations:** Department of Chemistry and Technology of Polymers, Faculty of Chemical Engineering and Technology, Cracow University of Technology, Warszawska 24, 31-155 Cracow, Poland

**Keywords:** transesterification, used cooking oil, catalyst, bio-polyol, polyurethane foam

## Abstract

The search for new sources of raw materials that can be used in the synthesis of polyurethanes and other polymer groups is extremely important. Currently, according to the principles of green chemistry and a circular economy, waste materials with a high reuse potential are being sought. This article presents a way of obtaining used-cooking-oil-based compounds capable of participating in the reaction of polyurethane creation. The transesterification reaction can be carried out using a variety of homo- and heterogeneous acid or base catalysts. Here, we looked at the impact of selected catalysts on the course of transesterification reactions, the composition of the post-reaction mixture and the possibility of using the products in polyurethane foam synthesis. The raw materials and the products were analyzed by means of gel permeation chromatography, FTIR spectroscopy and ^1^H NMR. The polyurethane foam formation process was analyzed using a FOAMAT^®^ apparatus.

## 1. Introduction

Two basic components are used for the synthesis of polyurethanes: polyhydric alcohols known as polyols and diisocyanates. Various methods of obtaining polyols from natural raw materials are described in the literature [1]. One of them is the transesterification of vegetable oils [2]. For this purpose, oils such as rapeseed oil, sunflower oil, palm oil and soybean oil can be used [3]. In addition, in recent years, the use of waste cooking oil as a raw material for the production of components applied in the synthesis of polyurethanes has attracted a lot of interest [2].

Transesterification is a generic name for one important class of reactions in organic chemistry in which one ester can be transformed into another via a partial alkoxy group exchange. In the process of the transesterification of vegetable oils, triglycerides (esters of fatty acids and glycerol) react with alcohol to form a mixture of fatty acid alkyl esters and glycerol.

The transesterification reaction is characterized by an equilibrium between the reactants [4]. The process is a sequence of three consecutive and reversible reactions in which the intermediates are di- and monoglycerides. According to the stoichiometry of the reaction, 1 mole of triglycerides reacts with 3 moles of alcohol. An excess of alcohol is used in practice to obtain higher yields of the products to allow for better phase separation of the esters and glycerol formed [5]. Redundant alcohol is typically used in biodiesel production, where a high degree of conversion of triglycerides to monoesters is required, and the unreacted transesterification agent is then removed from the mixture [6]. The effectiveness of the transesterification process of vegetable oils depends on many factors, including the type of catalyst, the molar ratio of alcohol to vegetable oil, the temperature and the purity of the reactants [7,8]. In contrast to the production of biodiesel, where monohydric alcohols, e.g., methanol [6], are used, at least a dihydric alcohol is required as a transesterification agent to obtain polyols. Most reports in the literature mention the use of glycerol, diethylene glycol and triethanolamine [9].

The transesterification reaction can be catalyzed by many types of catalysts having different reaction mechanisms. Literature reports on transesterification reactions most often describe the use of acid catalysts, such as sulfuric, sulfonic, phosphoric and hydrochloric acids [10,11]. This method is employable in various cases unless acid-sensitive components are involved [3]. An acid catalyst offers high efficiency, but the reaction rate is slow. Moreover, a higher oil/alcohol molar ratio and an extended reaction time are also required [11].

There are many reports on the use of base catalysts in the transesterification reaction. A wide variety of metal alkoxides are employable but the most popular ones are sodium and potassium alkoxides. Besides the alkoxides, metal acetate, oxides and carbonates are also used [10,12]. Alkali catalysts, such as sodium or potassium hydroxide, are considered the best catalysts for the transesterification reactions but they require a purification step since alkaline ions present in polyols catalyze side reactions (trimerization of isocyanates) during the foaming process [13]. These side reactions could affect the properties of the prepared polyurethane foam. In alkali-catalyzed transesterification, water can be formed in the reaction between the base and a transesterification agent. The presence of water and free fatty acid leads to soap formation via the hydrolysis of triglycerides, which can reduce the yield of the reaction and affect the quality of the product [11].

Solid catalysts have the potential to replace strong acids or bases to eliminate the corrosion problems of chemical equipment and the consequent environmental hazards posed by liquid acids/bases. A solid catalyst must have a porous structure with interconnecting pores so that the entire surface of the catalyst is available for promoting the transesterification reaction. The use of heterogeneous catalysts, i.e., ion exchange resins, allows for the elimination of product purification steps because such resins are easily removed from the reaction mixture via filtration or centrifugation. However, the disadvantage of such catalysts is the low temperature at which they should be used, which makes the transesterification reaction difficult to perform efficiently [14]. In the study of [14], a strongly acidic resin resistant to high temperature (Amberlyst 45) was used for biodiesel production via the methanolysis and ethanolysis of vegetable oil. The reaction resulted in a bio-diesel formation yield of more than 80%. In addition, as the researchers found, Amberlyst 45 can be used multiple times in the process. In the study conducted by [15], a polyol was obtained from palm oil and glycerol via transesterification using the following catalysts: p-toluenesulfonic acid, calcium hydroxide, aluminum hydroxide, sodium hydroxide and potassium hydroxide with two different concentrations of 0.5 and 2.0% (*w*/*w*) based on the weight of palm oil. The use of 2% NaOH as a catalyst in a reaction with a 6-molar excess of glycerol resulted in obtaining the best monoglyceride formation efficiency, i.e., about 78%.

There are also reports in the literature on the use of enzyme catalysts. Biocatalysts used in transesterification can be divided into three groups: free lipase, traditionally immobilized lipase (lipase immobilized on non-magnetic material) and lipase immobilized on magnetic nanoparticles [11]. Studies of transesterification catalyzed by lipase produced by *Candida antartica* [16,17,18], *Candida rugasa* and *Pseudomonas cepacia* [19] are described in the literature. The enzyme-catalyzed reaction requires a much longer time than the base-catalyzed reaction. While enzyme reactions are highly specific, the main problem of the lipase-catalyzed process is the high cost of lipases [16].

In industry, particular attention is paid to the economics of the process. It is preferred that the applied processes be single-stage and solvent-free, require no product purification, use waste materials and boast low energy consumption, i.e., should be conducted in the shortest time and at the lowest temperature. Therefore, an important issue is the selection of a suitable catalyst that will provide a high transesterification yield and its presence in the product will not interfere with the subsequent steps in the synthesis of a polyurethane. It is also important to select the catalyst for the appropriate transesterification agent. In the study presented here, transesterification reactions of used cooking oil were carried out using ethylene glycol and various catalysts. In our study, ethylene glycol, which is characterized by its low price, easy availability and low molecular weight, was used as a transesterification agent. Based on a literature review, several catalysts with different properties were selected and used for the synthesis of polyols via transesterification. The reaction products were analyzed in terms of their composition. Their potential to be used as bio-polyols in the synthesis of rigid polyurethane foams was also assessed.

## 2. Experimental Part

### 2.1. Materials

Used cooking oil (UCO) characterized by an iodine value of 104 gI_2_/100 g was supplied by local restaurants. Monoethylene glycol (MEG) was used as a transesterification agent and was supplied by Avantor Performance Materials Poland S.A (Gliwice, Poland). Nine different substances were used as catalysts: sodium hydroxide (NaOH), potassium hydroxide (KOH), aluminum hydroxide (Al(OH)_3_), calcium oxide (CaO), sulfuric acid (H_2_SO_4_) and potassium carbonate (K_2_CO_3_), which were supplied by Avantor Performance Materials Poland S.A. (Gliwice, Poland); zinc acetate ((CH_3_COO)_2_Zn), which was supplied by Honeywell Sp. z o.o. (Warsaw, Poland); and potassium acetate (CH_3_COOK) and tungstophosphoric acid (H_3_PW_12_O_40_), which were supplied by Merck Life Science Sp. z o.o. (Warsaw, Poland). Catalyst Polycat^®^ 9 was provided by Evonik, the surfactant Niax™ L-6900 was provided by Momentive Performance Materials GmbH Sp. z o.o. (Piaseczno, Poland) and polymeric methylene diphenyldiisocyanate (PMDI) with a free isocyanate groups content of 31 wt.% was supplied by Minova Ekochem S.A (Siemianowice Śląskie, Poland). Distilled water was also used to prepare the polyurethane foams.

### 2.2. Transesterification of the Waste Cooking Oil

The reaction was carried out in a 500 mL flask equipped with a magnetic stirrer set at 800 rpm, at atmospheric pressure, under reflux conditions and in an inert gas environment. The oil (250 g) was placed in the reactor and the temperature was set to 175 °C. After reaching the desired temperature, MEG with a selected catalyst was added to the reactor. MEG was added in an amount equal to the molar ratio of the oil and the transesterification agent of 1:3. The catalyst was added in an amount of 0.15% by weight relative to the total mass of the reactants. The reaction was carried out for 2 h at 175 °C. The reaction conditions used were based on previous research and a literature review [2,20,21].

### 2.3. Characterization of the Bio-Polyols

The hydroxyl value (Hval) was determined in accordance with the PN-C-89052-03:1993 standard. The acid value (Aval) was determined in accordance with PN-EN ISO 660:2009. The water content in polyols was determined in accordance with PN-81/C-04959 using the Karl Fisher method. The chemical structures of bio-polyols were analyzed using an FTIR spectrometer from Thermo Fisher Scientific, model Nicolet iS5 equipped with an ATR accessory and a diamond crystal. Spectra were recorded in the infrared range of 4000–500 cm^−1^. The number average molecular weight (*Mn*), weight average molecular weight (Mw) and dispersity (D) were determined using a gel permeation chromatography (GPC) analysis. GPC measurements were performed using a Knauer chromatograph equipped with a Plgel MIXED-E column and a refractometric detector.

The functionality (*f*) of the final bio-polyols was calculated following the formula below:$$f=\frac{Hval\cdot Mn}{56,110}$$

The chemical composition of each sample and the effect of the catalyst on the transesterification reaction were analyzed using ^1^H NMR analysis. Spectra were recorded with a Bruker Avance 500 MHz spectrometer (Billerica, MA, USA) with CDCl_3_ as the solvent at a temperature of 298 K. The tetramethylsilane (TMS) signal was used as a reference for the chemical shift values. The spectra were analyzed using MestReNova software (Mestrelab Research, S.L., Spain, Version 14.0.0).

### 2.4. Characterization of the Foaming Process

The bio-polyols obtained in the transesterification reaction were used to analyze the foaming process. The polyol premix (bio-polyol, catalyst, surfactant, water) was mixed for 30 s and then an isocyanate was added. Next, the whole reaction mixture was stirred for 5 s and poured into an open mold. The formulations of foams are summarized in Table 1. The weight of the isocyanate needed was calculated by taking into account the Hval of the resultant bio-polyols so that the value of the isocyanate index was equal to 1. 

An analysis of the polyurethane foaming process using a foam qualification system FOAMAT^®^ (Format Messtechnik GmbH, Karlsruhe, Germany) was carried out in a form made of cardboard. The foam qualification system FOAMAT^®^ consists of an ultrasonic sensor, a thermocouple, a pressure measurement and curing monitor devices. The measurement sequence and data processing are controlled using the ”FOAM ver.3.1” software. Dielectric polarization gives insight into the electrochemical processes that occur during foam formation. This parameter is essentially determined by chain-like molecules with a large dipole moment due to their polar ends (-OH, -NCO). Chain formation precedes the crosslinking reaction that ultimately suppresses all dipole mobility during the curing process. At the same time, it is possible to measure changes in the height and speed of growth of the expanding polyurethane system. 

## 3. Results and Discussion

The transesterification reactions were carried out in the presence of four groups of catalysts: acids, bases, salts and an oxide. The selected catalysts are among the most commonly used in transesterification reactions. Figure 1 shows photographs of the products of the used cooking oil transesterification with ethylene glycol using the following catalysts: H_2_SO_4_, H₃PW₁₂O₄₀, NaOH, KOH, Al(OH)_3_, CH_3_COOK, (CH_3_COO)_2_Zn, K_2_CO_3_ and CaO.

The labels assigned to the polyols include the type of raw material (UCO) and the type of catalyst that was used in the transesterification reaction. 

In the case of bio-polyols UCO_H_3_PW_12_O_40_ and UCO_Al(OH)_3_, separation of the products was observed, which may indicate a lack of or a low level of conversion of UCO in the reaction with MEG. The separation indicated that the mixture contained substances that did not mix and had different properties, i.e., polar (ethylene glycol) and non-polar (UCO). The remaining polyols were uniform liquids. Only a small amount of dark liquid (glycerol) or light sediment (saponification products) was observed at the bottom of the cylinder [22]. In addition, it was observed that the UCO_H_2_SO_4_ polyol had a much darker color compared with the others ones.

All transesterification reactions were performed at equivalent molar ratios. Neither byproducts nor catalysts were removed from the reaction mixture. This approach reduced the consumption of energy, water and chemicals by reducing the number of process steps. 

The transesterification reaction products were monitored through GPC, ^1^H NMR and FTIR analyses. 

The transesterification of UCO with MEG results in a product that is a mixture of different compounds. Figure 2 shows the theoretical course of the transesterification reaction of UCO with MEG [8,23,24,25]. 

The chemical composition of each sample and the effect of the catalyst on the transesterification reaction were analyzed using ^1^H NMR. 

The ^1^H NMR spectra of the UCO (Figure 3) revealed several typical signals of major groups of vegetable oils [26,27]: (**A**) δ (chemical shift) 0.86–0.90 ppm (–CH_2_–C**H_3_**), (**B**) δ 0.95–0.99 ppm (=CH–CH_2_–C**H_3_**), (**C**) δ 1.26–1.36 ppm ([–C**H_2_**–]_n_), (**D**) δ 1.61 ppm (β–CH_2_ group of the carbonyl group –C**H_2_**−CH_2_–OCO–), (**E**) δ 1.99–2.08 ppm (–C**H_2_**–CH=CH–), (**F**) δ 2.29–2.33 ppm (α–CH**_2_** group of the carbonyl group –CH_2_–C**H**_2_–CO–), (**G**) δ 2.76–2.82 ppm (=CH–C**H**_2_–CH=), (**H**) δ 4.13–4.31 ppm (methylene protons of glyceryl ROC**H**_2_–CH(OR′)–C**H_2_**OR″, (**I**) δ 5.24–5.30 ppm (methine protons of glyceryl ROCH_2_–C**H**(OR′)–CH_2_OR″), (**J**) δ 5.32–5.40 ppm (protons of olefin groups of fatty acids –C**H**=C**H**–) and (**K**) δ 3.71 ppm (methylene protons of ethylene glycol OH–C**H_2_**–C**H_2_**–OH).

The ^1^H NMR analysis confirmed that the resulting transesterification products were a mixture of many compounds, and their composition depended on the catalyst used. The presence of the MEG monoester and MEG diester; 1-/2-monoglycerides (1-MG/2-MG); 1,2-/1,3-diglycerides (1,2-DG/1,3-DG); unreacted substrates, i.e., triglycerides (TG) and MEG, as well as the glycerol formed in the reaction (Figure 2) were found. The samples tested were not purified or separated in any manner, which resulted in the appearance of some overlapping signals. The analysis of the signals was based on the studies of [26,27]. Figure 4 shows selected bands of the ^1^H NMR spectra of UCO and the transesterification products: (**L**) δ 5.24–5.30 ppm (glyceryl group in TG ROCH_2_-C**H**(OR′)-CH_2_OR″), (**M**) δ 5.07–5.11 ppm (glyceryl group in 1,2-DG ROCH_2_-C**H**(OR′)-CH_2_OH), (**N**) δ 4.89–4.93 ppm (glyceryl group in 2-MG HOCH_2_-C**H**(OR)-CH_2_OH), (**O**) [28] δ 4.27 ppm (methylene protons in ethylene glycol diester -O-C**H_2_**-C**H_2_**-O-), (**P**) δ 4.22–4.27 ppm (methylene protons of ethylene glycol monoester -O-C**H_2_**-CH_2_-OH), (**Q**) δ 4.22–4.27 ppm (glyceryl group in 1-MG ROC**H_2_**-CHOH-CH_2_OH), (**R**) δ 3.89–3.93 ppm glyceryl group in 1-MG ROCH_2_-CHO**H**-CH_2_OH), (**S**) δ 3.80–3.82 ppm (glyceryl group in 2-MG HOC**H_2_**-CH(OR)-C**H_2_**OH), (**T**) δ 3.76–3.72 ppm glyceryl group in 1,2DG ROCH_2_-CH(OR′)-C**H_2_**OH, (**U**) δ 3.71–3.66 ppm glyceryl group on 1-MG ROCH_2_-CHOH-**CH_2_**OH and (**V**) δ 3.59–3.69 ppm methylene protons of ethylene glycol monoester -CH_2_-C**H_2_**-OH).

The use of CaO and (CH_3_COO)_2_Zn catalysts led to the formation of MEG monoesters (signals P and V) without the production of MEG diesters (signal O). On the other hand, it was observed that significant amounts of 1,2DG (signal T) and small amounts of 1-MG and 2-MG (signals Q and S, respectively) were produced, indicating the incomplete conversion of UCO. Low-intensity peaks from the presence of MEG monoesters (signals P and V) were also observed for the H_2_SO_4_ catalyst.

The formation of ethylene glycol diesters (signal O), as well as high amounts of 1-MG and 2-MG (signals Q and S, respectively), were observed when H_2_SO_4_, NaOH, KOH, CH_3_COOK and K_2_CO_3_ were used as catalysts. However, signals from MEG monoesters (signals P and V) were not observed when using the abovementioned catalysts. The presence of unreactive ethylene glycol diesters in a bio-polyol is highly undesirable and significantly reduces the product’s suitability for use as a component in polyurethane formulations.

In the case of Al(OH)_3_ and H_3_PW_12_O_40_ as catalysts, no transesterification reaction was observed.

The products were also analyzed using GPC. The chromatograms of the products and data are shown in Figure 5 and Table 2, whereas Table 3 presents their selected properties.

GPC separates based on the hydrodynamic volume of the analytes. If the analyzed compounds have approximately similar hydrodynamic volumes, they will appear as one peak in a chromatogram, as they elute from the chromatograph column at a similar time. The separation resolution of the chromatograph that was used does not allow for the separation of a mixture consisting of monoglycerides and MEG monoesters, nor diglycerides and MEG diesters, because they have very similar hydrodynamic volumes and molar masses [23].

The chromatograms show peaks coming from triglycerides (peak 1), diglycerides or MEG diesters or a mixture of these two compounds (peak 2), monoglycerides and MEG monoesters or a mixture of these two compounds (peak 3), and unreacted MEG and glycerol (peak 4). It was observed that the lowest yield of the reaction was obtained with the use of the catalysts Al(OH)_3_ and H_3_PW_12_O_40_, which was confirmed by the highest content of triglyceride (peak 1). The content of triglyceride in these products was about 87%, whereas only negligible amounts of di- and monoglyceride/esters were present. This showed that the transesterification reaction catalyzed by H_3_PW_12_O_40_ and Al(OH)_3_ was almost completely absent and the catalysts were ineffective under the conditions of the reactions. It can be concluded that the two layers present in these products (Figure 1) were the UCO layer and the unreacted MEG layer. 

The products obtained as a result of transesterification carried out with the use of the catalysts containing potassium and sodium ions (strongly basic), i.e., NaOH, KOH, K_2_CO_3_ and CH_3_COOK, were characterized using the highest contents of monoglycerides/MEG monoesters desirable as components for the production of polyols. These products were characterized by a monoglycerides/MEG monoesters content of over 50 wt.% and a content of unreacted UCO below 2 wt.%. The bio-polyols UCO_H_2_SO_4_, UCO_CaO and UCO_(CH_3_COO)_2_Zn were characterized by a monoglycerides/MEG monoesters content of about 40 wt.%. From the values of reaction rate constants reported in the literature, it follows that reactions catalyzed by alkali catalysts occur several hundred or even several thousand times faster than those catalyzed by acids. Therefore, higher triglyceride conversion rates were obtained with alkali catalysts in the case of the experiments conducted. In their case, the step that determines the speed of the reaction is the last step, i.e., the transition from monoglyceride to glycerol [22,25].

One of the most important properties of bio-polyols is their Hval, which determines the application of a particular polyol and its effect on the properties of polyurethane foams. The values of Hval, Aval, Mn and Mw dispersity, and functionality are presented in Table 3. The polyols UCO_ KOH, UCO_K_2_CO_3_, UCO_CH_3_COOK and UCO_NaOH had the highest values of Hval with 281, 280, 280 and 269 mgKOH/g, respectively. All polyols had Aval values below 1.0 mgKOH g, with the exception of the polyol catalyzed with sulfuric acid (5.55 mg KOH/g and a low content of water (<0.05 wt.%).

FTIR analyses of the products were also performed (Figure 6). The FTIR spectra of the polyols synthesized in our experiment correlated with the GPC results, as well as the hydroxyl numbers. The broad band at 3340–3390 cm^−1^ corresponded to free hydroxyl groups in the bio-polyols. The region of hydroxyl group vibration is shown in detail in Figure 6a. The absorption intensities were the highest for UCO_KOH, UCO_NaOH and UCO_K_2_CO_3_. The lower the hydroxyl number, the lower the absorption intensity. For UCO, there was no peak in this area, while for the transesterification products catalyzed with H_3_PW_12_O_40_ and Al(OH)_3_, the peak in this area originated from the unreacted MEG and had a very low intensity.

In the 3000–2800 cm^−1^ range (Figure 6b), two intense bands at 2920 and 2850 cm^−1^ were observed due to stretching vibrations of the CH_2_ groups of the aliphatic structures in the chain backbone [29]. An absorption band at 1735–1743 cm^−1^ corresponded to the carbonyl C=O bonds in the ester groups (Figure 6c). That peak was the highest for the unreacted UCO, UCO_H_3_PW_12_O_40_ and UCO_AL(OH)_3_. As the transesterification reaction progressed toward monoglycerides, the intensity of this peak decreased.

Foaming a polyurethane system is the most important stage during polyurethane (PUR) foam preparation. In this process, the reaction mixture increases its volume several dozen times. During the foaming process, both foaming and gelling reactions take place in parallel, and thus, each component added to the polyurethane system affects the foaming process. The chemical structure of bio-polyol, its Hval and the presence of additional groups in its structure, e.g., an amine group, have an impact on the foaming process, as shown in previous studies [30]. Given the fact that the reaction to obtain bio-polyols (with characteristics presented in Table 3) is a single-stage process and the catalyst is not removed from the reaction medium, it is important to analyze the effect of the polyols obtained with various catalysts on the foaming process.

In order to analyze the foaming process of PUR systems, seven different bio-polyols were selected. Figure 7 shows the effect of the bio-polyols obtained via transesterification with the use of various catalysts ((CH_3_COO)_2_Zn, CH_3_COOK, NaOH, KOH, CaO, K_2_CO_3_, H_2_SO_4_) on the course of the foaming process of polyurethane systems, including the temperature in the foam core (a), change in dielectric polarization (b), pressure (c) and foam height (d). The course of the foaming process was analyzed for a period of 600 s. The formulations of the PUR systems are given in Table 1. Table 4 summarizes the parameters of the foaming process, i.e., characteristic times for the formation of a foam, maximum temperature and pressure, and shrinkage.

It was observed that when using different bio-polyols, different temperatures were achieved in the foam core. The highest temperatures, reaching approx. 150 °C, were found in the materials obtained with the following bio-polyols: UCO_KOH, UCO_NaOH, UCO_CH_3_COOK and UCO_K_2_CO_3_ (Figure 7a). The lowest temperature of 133 °C was inside the PUR foam obtained with the use of bio-polyol catalyzed with CaO. This bio-polyol had the lowest hydroxyl number and the lowest monoglyceride content.

On the basis of the dielectric polarization measurements (Figure 7b), it was found that the PUR systems obtained using the bio-polyols synthesized with (CH_3_COO)_2_Zn and CH_3_COOK were characterized by the highest reactivity. In this case, the dielectric polarization reached approx. 10% of its initial value the fastest, after about 300 s. A dielectric polarization value close to 0 indicates that the reactions took place and there was a small amount of free hydroxyl and isocyanate groups in the mixture. The systems with the bio-polyols UCO_K_2_CO_3_ and UCO_KOH were also very reactive. The systems containing UCO_MEG_H_2_SO_4_ and UCO_CaO were characterized by the slowest decrease in dielectric polarization, and thus, in reactivity.

The pressure exerted by the forming foam on the measuring table was also analyzed. The highest pressures were in the case of the foams obtained with the bio-polyols UCO_NaOH, UCO_KOH and UCO_K_2_CO_3_ (Figure 7c). The foams that reached high pressures also had high shrinkage values (Table 4, Figure 8). 

The above-described analyses show that the transesterification carried out with the use of reactive catalysts such as NaOH or KOH led to the formation of side products, namely, MEG diesters, that do not have hydroxyl groups capable of reacting with isocyanates. The bio-polyols contained different amounts of MEG and glycol, which are small molecules compared with the mono- and diglycerides/esters that are formed in the reaction. Consequently, MEG and glycol have a greater potential to react with isocyanate because there are fewer steric hindrances between these components. MEG and glycol, which are involved in the formation of polyurethane bonds, cause the reaction to produce a polyurethane foam with a greater number of rigid segments. 

Figure 8 shows a cross-section of the foams in a cardboard tube 24 h after their synthesis. The foams PU_UCO_H_2_SO_4_, PU_UCO_CaO and PU_UCO_(CH_3_COO)_2_Zn had the highest dimensional stability and adhered well to the cardboard tube wall. The catalysts used to obtain those foams led to the formation of MEG-based monoesters and diglycerides. All other foams, which contained bio-polyols characterized by the presence of MEG diesters (non-reactive compounds) in the absence of MEG monoesters and a higher amount of monoglycerides, were not dimensionally stable and exhibited significant shrinkage, especially the PU_UCO_CH_3_COOK foam. 

## 4. Conclusions

In the conducted research, transesterification reactions were carried out with the use of selected catalysts. Depending on the catalyst used, a different composition of the final mixture was obtained. The following products were identified in the final mixture: glycerol, MEG, triglyceride, diglyceride, monoglyceride, MEG diesters and an MEG monoester.

Two of the selected catalysts worked very poorly, as evidenced by the low proportion of di- and monoglycerides/MEG esters in the post-reaction mixture. Those were the catalyst H_3_PW_12_O_40_, for which the amount of monoglyceride/MEG monoester was 1.14%, and Al(OH)_3_, with 0.37% of monoglyceride/MEG monoester in the final product mixture. The use of those two catalysts in amounts greater than 0.15% by weight and their effectiveness at higher temperatures than 175 °C or extended testing times above 2 h were not studied because such procedures are not compatible with the principles of green chemistry. 

The efficiency of monoglyceride/ethylene glycol monoester formation was between 39 and 53 wt.% for the reactions catalyzed by sulfuric acid and potassium hydroxide, respectively. The products obtained as a result of transesterification carried out with the use of the catalysts containing potassium and sodium ions (strongly basic), i.e., NaOH, KOH, K_2_CO_3_ and CH_3_COOK, were characterized by the highest contents of monoglycerides. The products were characterized by a monoglyceride content of over 50% and a content of unreacted used cooking oil below 2%. However, those compounds also catalyzed the formation of MEG diesters. A MEG diester does not contain reactive hydroxyl groups that could participate in the polyurethane bond formation reaction; therefore, they are not desirable products.

The use of H_2_SO_4_, CaO and (CH_3_COO)_2_Zn as catalysts resulted in products containing ethylene glycol monoesters. No ethylene glycol diesters were observed in these cases. The products obtained using these catalysts were characterized by the presence of monoglyceride/ethylene glycol monoester content of about 40 wt.%. The resultant bio-polyols had hydroxyl values in the range of 170–280 mg KOH/g. Rigid polyurethane foams were obtained from the synthesized polyols. 

The catalyst (CH_3_COO)_2_Zn can be considered the best choice. The bio-polyol obtained with it was characterized by both higher hydroxyl value and higher ethylene glycol monoglycerol/monoester content compared with the bio-polyol obtained in the CaO-catalyzed reaction. The polyurethane system obtained using the bio-polyol obtained in the (CH_3_COO)_2_Zn-catalyzed reaction was also characterized by the fastest start time (highest reactivity) and lowest shrinkage, as well as the best adhesion to the test cardboard tube 24 h after manufacture.

## Figures and Tables

**Figure 1 materials-15-07807-f001:**
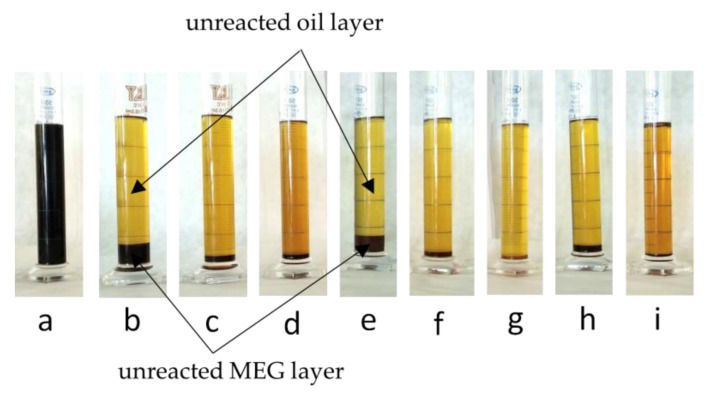
Bio-polyols obtained using the following catalysts: (**a**) H_2_SO_4_, (**b**) H_3_PW_12_O_40_, (**c**) NaOH, (**d**) KOH, (**e**) Al(OH)_3_, (**f**) CH_3_COOK, (**g**) (CH_3_COO)_2_Zn, (**h**) K_2_CO_3_ and (**i**) CaO.

**Figure 2 materials-15-07807-f002:**
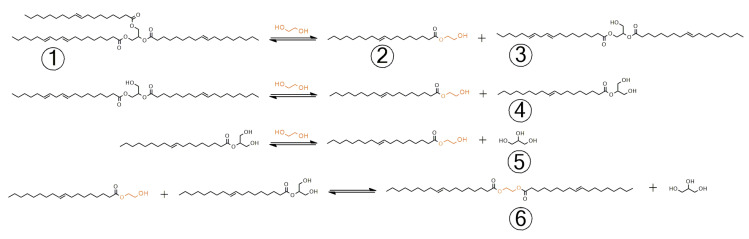
Transesterification course and the reaction products: 1—triglyceride, 2—ethylene glycol monoester, 3—diglyceride, 4—monoglyceride, 5—glycerol and 6—ethylene glycol diester.

**Figure 3 materials-15-07807-f003:**
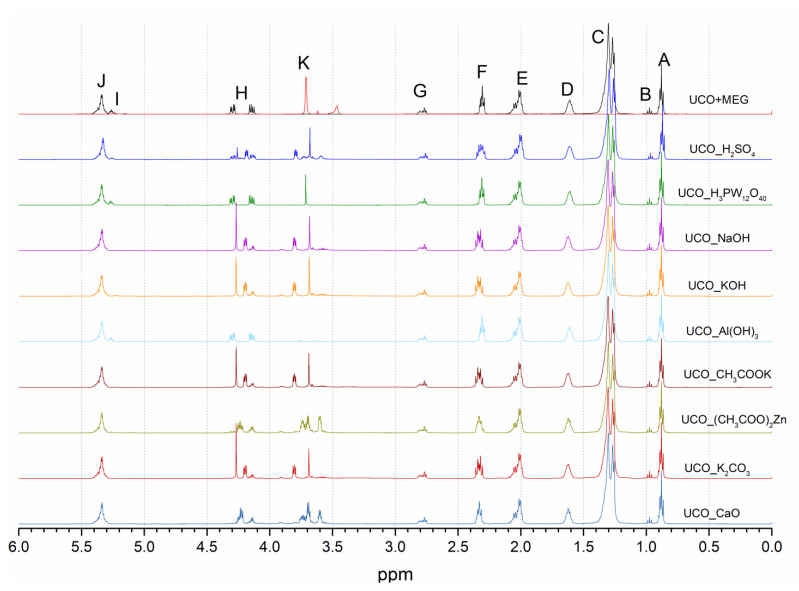
Overall ^1^H NMR spectra of the UCO and polyols.

**Figure 4 materials-15-07807-f004:**
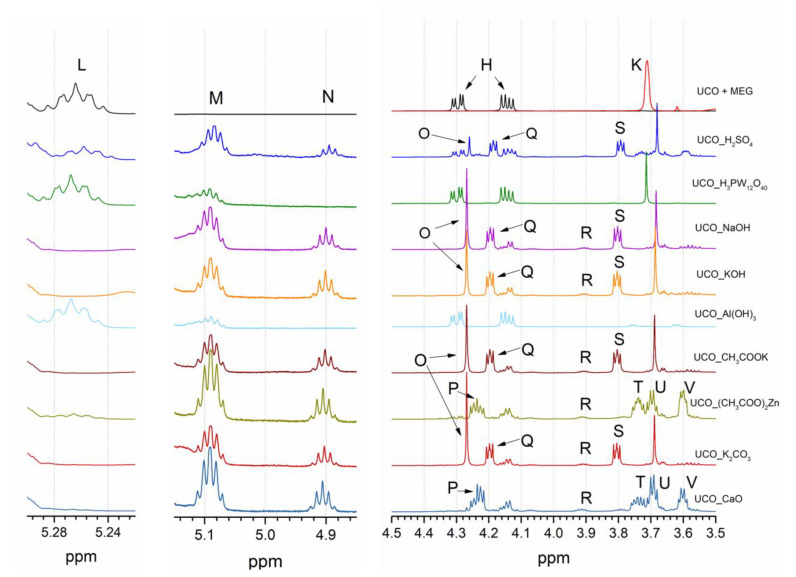
^1^H NMR spectra of the UCO and polyols in specific chemical shift ranges.

**Figure 5 materials-15-07807-f005:**
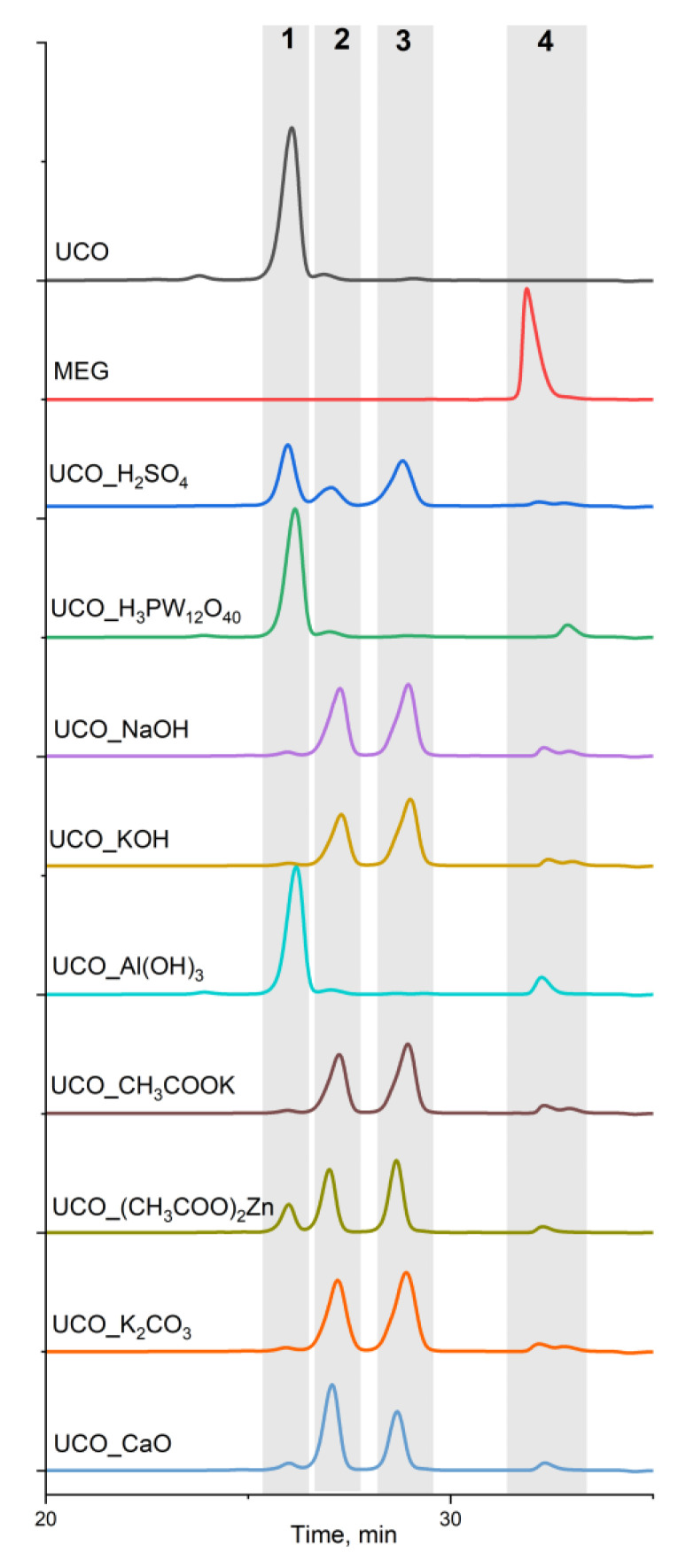
Chromatograms of the reaction products, 1—triglycerides, 2—diglycerides/ MEG diesters, 3—monoglycerides/ MEG monoesters, and 4—unreacted MEG and glycerol.

**Figure 6 materials-15-07807-f006:**
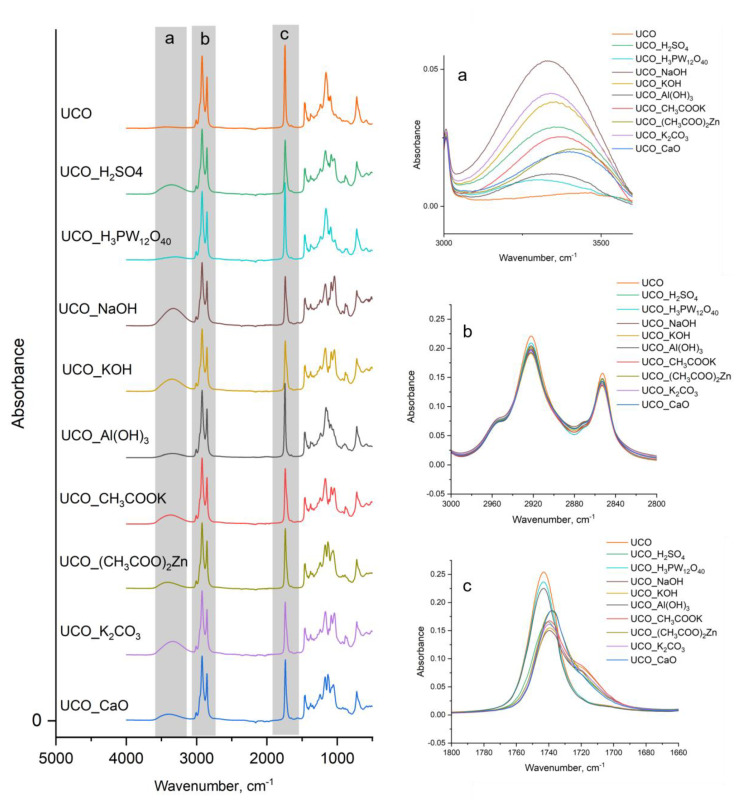
FTIR spectra of the bio-polyols. (**a**) the region of hydroxyl group vibration, (**b**) the stretching vibrations of the CH_2_ groups of the aliphatic structures in the chain backbone, (**c**) the absorption band of carbonyl C=O bonds in the ester groups.

**Figure 7 materials-15-07807-f007:**
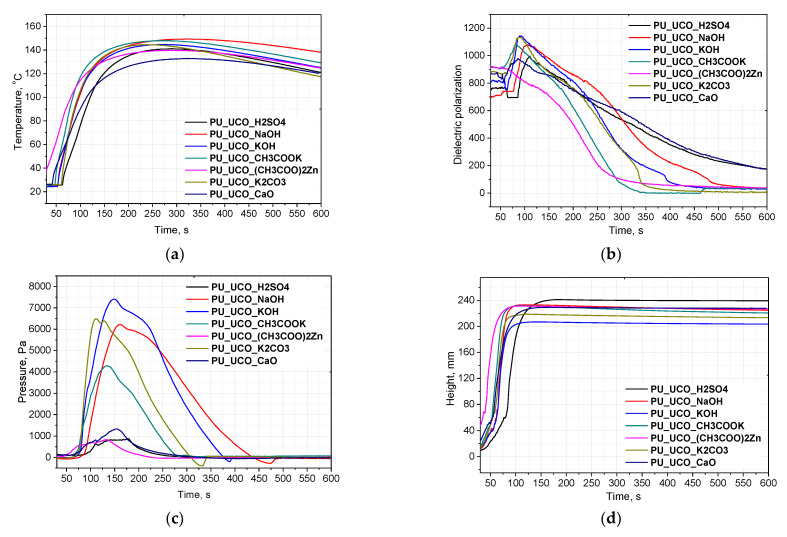
Temperature in the foam core (**a**), change in dielectric polarization (**b**), pressure (**c**) and foam height (**d**).

**Figure 8 materials-15-07807-f008:**
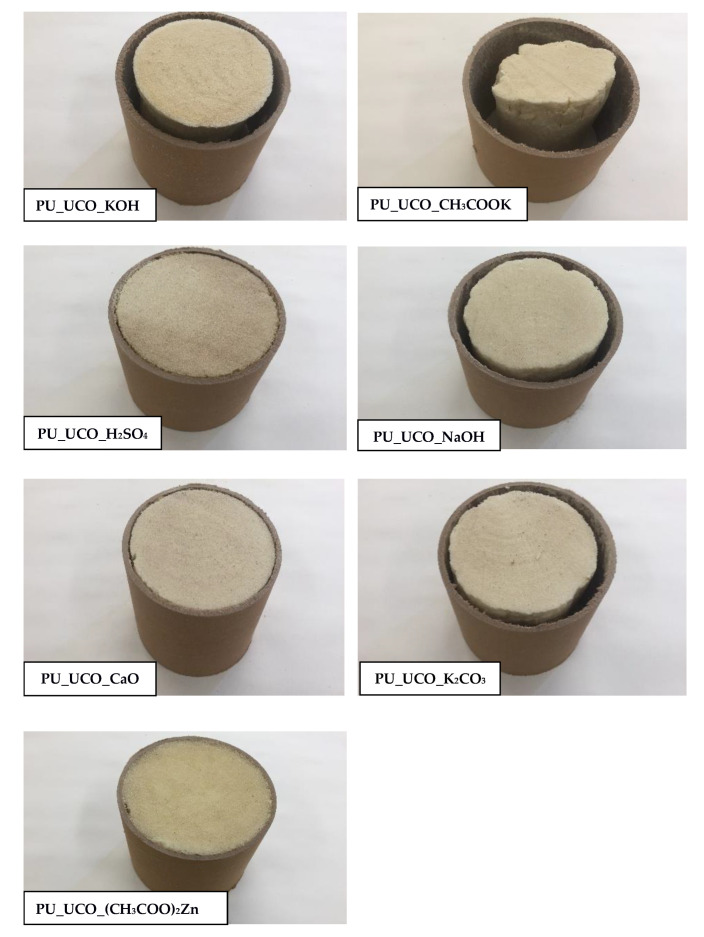
Cross-sections of polyurethane foams obtained with various bio-polyols.

**Table 1 materials-15-07807-t001:** Formulations of the foams.

Component, g	UCO_H_2_SO_4_	UCO_NaOH	UCO_KOH	UCO_CH_3_COOK	UCO_(CH_3_COO)_2_Zn	UCO_K_2_CO_3_	UCO_CaO
Bio-polyol	100	100	100	100	100	100	100
Polycat^®^ 9	1.0	1.0	1.0	1.0	1.0	1.0	1.0
Niax™ L-6900	1.5	1.5	1.5	1.5	1.5	1.5	1.5
H_2_O	3.0	3.0	3.0	3.0	3.0	3.0	3.0
Isocyanate	93.4	110.0	112.9	112.7	99.4	112.5	86.2

**Table 2 materials-15-07807-t002:** Effect of catalysts on the chemical compositions of bio-polyols obtained in the synthesis: 1—triglycerides, 2—diglycerides/diesters, 3—monoglycerides/ MEG monoesters, and 4—unreacted MEG and glycerol.

	Peak Area, %			
	1	2	3	4
UCO_H_2_SO_4_	39.87	15.39	39.41	5.33
UCO_H_3_PW_12_O_40_	87.86	3.88	1.14	7.12
UCO_NaOH	1.43	44.85	50.19	3.52
UCO_KOH	1.59	38.53	53.65	6.23
UCO_Al(OH)_3_	87.68	1.82	0.37	10.12
UCO_CH_3_COOK	1.57	40.75	50.61	7.06
UCO_(CH_3_COO)_2_Zn	15.66	37.66	43.04	3.64
UCO_K_2_CO_3_	1.69	41.95	50.13	6.24
UCO_CaO	2.78	54.46	38.09	4.67

**Table 3 materials-15-07807-t003:** Properties of bio-polyols obtained via reactions with various catalysts.

	Bio-Polyol Properties						
	Hval, mgKOH/g	Aval, mgKOH/g	Mn, g/mol	Mw, g/mol	f	D	H_2_O%, wt.%.
UCO_H_2_SO_4_	199.6	5.55	466	637	1.7	1.37	0.42
UCO_H_3_PW_12_O_40_	-	-	482	792	-	1.64	-
UCO_NaOH	269.3	0.43	393	484	1.8	1.23	0.04
UCO_KOH	281.2	0.44	374	460	1.9	1.23	0.05
UCO_Al(OH)_3_	-	-	539	802	-	1.49	-
UCO_CH_3_COOK	280.4	0.63	403	480	1.5	1.19	0.03
(CH_3_COO)_2_Zn	224.7	0.82	455	574	2.0	1.26	0.04
UCO_K_2_CO_3_	279.8	0.20	380	475	1,9	1,25	0,02
UCO_CaO	169.0	0.41	465	557	1.5	1.20	0.03

Hval—hydroxyl number; Aval—acid number; Mn—number average molecular weight; Mw—weight average molecular weight; D—dispersity; H_2_O%—water content.

**Table 4 materials-15-07807-t004:** The most important properties of the foaming process of polyurethane materials.

	Properties of the Foaming Process				
Catalyst	Start Time, s	Rise Time, s	Max. Temperature, °C	Max. Pressure, Pa	Shrinkage, %
H_2_SO_4_	75 ± 8	146 ± 2	140 ± 2	800 ± 100	0.7 ± 0.1
NaOH	51 ± 1	99 ± 10	149 ± 1	7400 ± 1600	2.9 ± 1.1
KOH	57 ± 2	98 ± 3	150 ± 7	7000 ± 550	2.3 ± 0.9
CH_3_COOK	33 ± 1	86 ± 1	147 ± 1	4500 ± 350	4.7 ± 0.6
(CH_3_COO)_2_Zn	19 ± 2	80 ± 2	139 ± 1	900 ± 50	2.4 ± 0.2
K_2_CO_3_	43 ± 2	97 ± 11	147 ± 4	7000 ± 700	2.4 ± 0.3
CaO	35 ± 7	120 ± 1	133 ± 0	1500 ± 200	0.5 ± 0.1

## Data Availability

Not applicable.

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
