# Peer review of "Impact of Various Catalysts on Transesterification of Used Cooking Oil and Foaming Processes of Polyurethane Systems"

_materials, 2022, doi:10.3390/ma15217807_

Round 1
Reviewer 1 Report
The purpose and novelty of the paper are clearly explained. However, there are several questions and suggestions that need to be addressed:
1. Kindly add explanations on the reasons for the chosen experimental reaction condition, such as temperature, catalyst weight, and reaction time.
2. On P5L186, the low-level separation of UCO_H3PW12O40 and UCO_Al(OH)3 are poorly observed based on Figure 1.
3. The explanation of 1H NMR and FT-IR results is clearly discussed.
4. Kindly input references on the included theory. For example on P8L248. It is said that "The transesterification reaction is a reversible reaction. The formation of ethylene glycol diesters (signal O), which do not contain hydroxyl groups (and consequently are not subject to further reactions), leads to a shift of the transesterification reaction towards the products." which needs further justification based on the reference paper.
Author Response
Dear Rewiver,
Thank you very much for the submitted review and comments on the article. We responded to all comments in the answer below. The changes were introduced in the manuscript and marked in color.
- Kindly add explanations on the reasons for the chosen experimental reaction condition, such as temperature, catalyst weight, and reaction time.
Reaction conditions result from previous research and literature review. Information about this has been added to the manuscript.
- On P5L186, the low-level separation of UCO_H3PW12O40and UCO_Al(OH)3 are poorly observed based on Figure 1.
Figure 1 shows a photo of the reaction products placed in measuring cylinders. In the case of these catalysts (H3PW12O40 and Al(OH)3 ), a larger bottom layer can be observed. The volumes of these layers correspond approximately to the volumes of the reactants dosed for the reaction. The composition of these mixtures was also confirmed by GPC analysis and the analysis of the hydroxyl value. In other polyols, the bottom layer is much smaller. This layer consists of glycerin and unreacted MEG. Descriptions and arrows have been added to identify the layers in the Fig. 1.
- Kindly input references on the included theory. For example on P8L248. It is said that "The transesterification reaction is a reversible reaction. The formation of ethylene glycol diesters (signal O), which do not contain hydroxyl groups (and consequently are not subject to further reactions), leads to a shift of the transesterification reaction towards the products." which needs further justification based on the reference paper.
The selected fragment was reanalyzed. The conclusions of the 1H NMR analysis have been reformulated. Where possible, literature was added. However, most publications describe the reaction and kinetics during the biodiesel production process, in which monofunctional alcohols are used, therefore there is no possibility of the formation of diesters as co-products.
Reviewer 2 Report
The manuscript (materials-2000579) evaluated different catalysts for the transesterification reaction. The study is interesting and will be valuable for the further catalyst selecting in the catalytic transformation of the used oil. To be more publishable, the following comments should be addressed.
Comment 1. Please make clarification of the mechanism for the transesterification reaction. Which is the rate-determing step and what is the requirement for the catalyst?
Comment 2. Although the author has compared different catalyst for the catalytic reaction, however, I can not see a general structure-function relationship and a general information of which catalyst is the best choice.
Comment 3. Please make the cyclability and stability of the catalyst into consideration, which is very important in the practical applications.
Comment 4. Please pay attention to the format of the Figures and the caption, e.g. Fig. 3. Please check the whole manuscript and make corresponding exaplanations.
Comment 5. The following manuscripts ares suggested for the citation. Applied Energy 87 (2010) 1083–1095, Chem. Soc. Rev., 2008, 37, 527–549, Applied Surface Science, 2022, 600, 15404
Author Response
Dear Reviewer,
Thank you very much for the submitted review and comments on the article. We responded to all comments in the answer below. The changes were introduced in the manuscript and marked in color.
Comment 1. Please make clarification of the mechanism for the transesterification reaction. Which is the rate-determing step and what is the requirement for the catalyst?
Transesterification is a reversible reaction, which means it can proceed in two directions. An equilibrium is established between the substrates. In order to achieve a high degree of triglyceride conversion, one of the reactants (alcohol) should be used in excess. [Noureddini H. and Zhu D., JAOCS, 74, 11, 1997], [Freedman, B., JAOCS, 63,10, 1986, 1375]. As mentioned in the article, many types of catalysts can be used in the transesterification reaction, i.e. acidic, basic, non-ionic, hetorogenic and enzymatic. In the study, no excess glycol was used, only a stoichiometric ratio, so the yield of monoester/monoglyceride formation was obtained up to 53%. The stage that determines the rate of reaction is the slowest stage. The literature shows that for reactions catalyzed by an acid catalyst, the slowest step is the step of forming diglyceride from triglyceride. In contrast, for an alkaline catalyst, the decisive step will be the transition from monoglyceride to glycerol. In addition, the reaction rate constants for alkaline catalysts are several hundred times higher than for acid catalysts. [Freedman B. at al, JAOCS, 63, 10, 1986] This information has been added to the manuscript.
Comment 2. Although the author has compared different catalyst for the catalytic reaction, however, I can not see a general structure-function relationship and a general information of which catalyst is the best choice.
Based on our studies, we observed that strong bases (NaOH, KOH) and salts of basic character (CH3COOK) and K2CO3 catalyzed the reaction to form ethylene glycol diesters, which do not contain the hydroxyl groups necessary for the reaction with isocyanate. Strongly acidic sulfuric acid (VI) also led to the formation of diesters, but in smaller amounts than with base catalysts, due to the lower catalytic activity of acids in the transesterification reaction compared to bases. In the case of basic oxide (CaO) and (CH3COO)2Zn which acts as a Lewis acid, the formation of diesters of ethylene glycol and fatty acids was not observed. The behavior of the mentioned reactants was clearly confirmed by 1H NMR analysis.
The catalyst (CH3COO)2Zn can be considered as the best choice. The bio-polyol obtained with it was characterized by both higher hydroxyl value and higher ethylene glycol monoglycerol/monoester content compared to the bio-polyol obtained in the CaO-catalyzed reaction. The polyurethane system obtained using the bio-polyol obtained in the (CH3COO)2Zn-catalyzed reaction was also characterized by the fastest start time (highest reactivity) and lowest shrinkage, as well as the best adhesion to the test cardboard tube 24 h after manufacture.
Comment 3. Please make the cyclability and stability of the catalyst into consideration, which is very important in the practical applications.
The cyclicity and stability of the catalyst is important, but in the conducted studies the catalysts used were not separated after the synthesis process and remained in the product. Often the catalyst recovery step is costly and environmentally unfriendly. Therefore, in this work, the catalysts used remained in the reaction mixture, and in further stages of the study, it was checked whether their presence affected the process of polyurethane foam formation.
Comment 4. Please pay attention to the format of the Figures and the caption, e.g. Fig. 3. Please check the whole manuscript and make corresponding explanations.
Figure 3 has been revised and the remaining figures have been checked.
Comment 5. The following manuscripts are suggested for the citation. Applied Energy 87 (2010) 1083–1095, Applied Surface Science, 2022, 600, 15404
Citations have been added. One of the articles indicated by the reviewer was cited. Unfortunately, the other article could not be found.